# Initial Description of Pilotage and Tug Services in the Context of e-Navigation

## Adam Weintrit 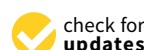

Department of Navigation, Gdynia Maritime University, 81-345 Gdynia, Poland; a.weintrit@wn.umg.edu.pl

**Abstract:** The International Maritime Organization (IMO)'s Maritime Safety Committee (MSC), at its 101st session (5 to 14 June 2019), adopted Resolution MSC.467(101) on the guidance on the definition and harmonization of the format and structure of maritime services in the context of e-Navigation and agreed to consolidate the descriptions of maritime services and to consider them together with all involved international organizations and interested member states, in order to harmonize the provision and exchange of maritime information and data. In doing so, the MSC also approved the initial descriptions of maritime services in the context of e-Navigation (IMO MSC.1/Circ.1610), which had been prepared by the Sub-Committee on Navigation, Communications and Search and Rescue, at its sixth session (16 to 25 January 2019). The information contained in this paper constitutes the descriptions of two selected examples of maritime services, an initial contribution for the harmonization of the formats and structures of pilotage and tug services. The initial description of each of maritime services is expected to be next periodically updated, taking into account developments and related work on international harmonization.

**Keywords:** marine navigation; safety at sea; maritime services; e-Navigation; IMO; MSC; NCSR; pilotage service; tug service

## 1. Background

The present situation in international shipping is characterized by fast technological development, affecting basic concepts of ship operation and even changing traditional paradigms of ship control. The e-Navigation concept developed by the International Maritime Organization (IMO) and the International Association of Marine Aids to Navigation and Lighthouse Authorities (IALA) focuses on better and more comprehensive support of human operators. However, modern information and communication technologies (ICT) not only are the core to the implementation of the e-Navigation strategy, but provide good foundation for automation of systems. The progressing digitalization further presses ahead application of integrated and automated systems to steer even large sea-going ships. The manifold abilities of these technologies and companies looking for more cost-effective solutions present autonomous navigation and unmanned shipping as soon to come. Taking this for granted, it will not happen that all ships will be unmanned and autonomous immediately. It is assumed that there will be a transition period, not necessarily short, in which unmanned vessels will operate together with either unmanned autonomous or unmanned remote-controlled vessels. Mixed traffic scenarios appear to be particularly difficult in terms of ship traffic safety and efficiency [1,2].

## 2. Introduction

The majority of accidents in the maritime domain are caused by human errors. One of the measures to reduce human-related marine accidents was proposed to implement the e-Navigation concept according to the IMO strategy implementation plan (SIP) [3–5]. e-Navigation is defined as the

"harmonized collection, integration, exchange, presentation and analysis of marine information on board and ashore by electronic means to enhance berth to berth navigation and related services for safety and security at sea and protection of the marine environment". To achieve this, the introduction of electronic means, including state-of-the-art ICT, is the key to support ship operators on board as well as ashore. Considering the fact that around 50% of accidents occurring at sea are attributed to navigational challenges, a systematic maritime traffic management seems to be necessary. On the other hand, modern ship operations rely on a small number of crew, whose responsibilities for safe and efficient navigation are increasingly high. Without operational support from the shore, using a reliable technology-based system, it would be challenging to reduce marine accidents. e-Navigation provides a great potential to help mitigate incidents such as collisions [6], grounding problems, oil spills, and piracy. It also allows, for certain, utilizing both new and existing technologies, which are acceptable within operating standards.

The Maritime Safety Committee (MSC), at its 101st session (5 to 14 June 2019), approved the guidance on the definition and harmonization of the formats and structures of maritime services (MSs) in the context of e-Navigation [7] and agreed to consolidate the descriptions of MSs and to consider them together with all involved international organizations and interested member states, in order to harmonize the provision and exchange of maritime information and data. In doing so, the MSC also approved the initial descriptions of MSs in the context of e-Navigation, which had been prepared by the Sub-Committee on Navigation, Communications and Search and Rescue, at its sixth session (16 to 25 January 2019) [8]. The author was a member of the IMO expert group that developed documents regarding the guidance on the definition and harmonization of the formats and structures of MSs in the context of e-Navigation (IMO MSC.467(101) [7]). He was also involved in the IMO MSC.1/Circ.1610 [8] regarding the initial descriptions of MSs in the context of e-Navigation.

The present times have witnessed fast technology development/implementation in the shipping industry. Concepts like e-Navigation are focused on better and more comprehensive support of human operators. Automation, modernizing information, and communication systems are the future in ship operation. Several companies are already talking about unmanned and autonomous ships. These technologies developed in shipping industry contribute to safer, more efficient and sustainable operations at sea, both in containers and human transportation.

Considering that most maritime accidents are due to human errors, one of the actions is to implement e-Navigation to reduce human-based sea accidents. e-Navigation intends to enhance navigation and related services for safety and security at sea and protection of the marine environment. The development of new technologies and their implementation for e-Navigation can implement systematic maritime traffic management and therefore reduce marine accidents.

In this paper, the author studied the operational use of the proposed e-Navigation concept in terms of its requirements and implementation plans as well as potential limitations and benefits around the concept in selected sectors. The information contained in this paper constitutes the initial descriptions of two chosen examples of MSs, an initial contribution for the harmonization of the formats and structures of pilotage and tug services. The initial description of each of MSs is expected to be next periodically updated, taking into account developments and related work on international harmonization.

The majority of the paper is intentionally directly prescribed from the IMO MSC 1/Circ. 1610 [8]. Although this document is referenced several times in the text, which parts of the text are actually direct copies is not clear. The author decided that this must be clarified. Therefore, it was done graphically by using italics in the text. It should be clear that the author took part in writing the initial IMO document and the IMO documents that Circ. 1610 are based on. The development of a maritime service portfolio (MSP) is necessary for the success of e-Navigation. However, one may ask why the author thought that it was necessary to write a paper supporting something that had already been published by the IMO. The main reasons are political, which make such academic support useful and of course have to be confirmed.

## 3. MSPs

An MSP in the context of e-Navigation can be defined as a set of operational MSs and associated technical services provided in a unified, digital format. Hence, an MSP may also be construed as a set of "products" provided by a stakeholder in a given sea area, waterway, or port as appropriate. Such marine information has been termed "maritime services", and this guidance envisages harmonizing the structures and formats of digitally transmitted data and information and to display them in a harmonized way on a ship's bridge or shore-based facilities broadcasting and receiving marine information.

### 3.1. MSP Definition

Before the work on the development on the definition and harmonization of the formats and structures of MSPs commenced, the IMO Secretariat considered that a clear understanding of MSPs is indispensable. A definition of MSP can be found in [9,10], that is, an MSP defines and describes a set of operational and technical services and levels of services provided by a stakeholder in a given sea area, waterway, or port as appropriate.

The SIP [7] identified 16 MSPs, including the type of service provided by each MSP, as well as the associated responsible service provider. It is evident that the services (MSPs) vary significantly, ranging from, for example, vessel traffic service (VTS) information to a ship, medical information and instructions provided by doctors to the ship's crew responsible for medical care to ice navigation, route information, search and rescue coordinates and many more. The sets of data, instructions, and information are very different in nature and could take numerical values, geographical coordinates, medical terminology, courses to steer, waypoint coordinates, communication channels, and many more.

As outlined in later IMO documents regarding the e-Navigation output on harmonized MSPs, MSPs are considered to form a framework for the electronic provision of information related to MSs in a harmonized way between the shore and ships. It is therefore necessary to harmonize the format, structure, and communication channels used to exchange [11,12]. It was also argued that a lack of coordination in the provision of information related to MSs among organizations responsible for the provision of MSPs may lead to duplication of efforts, development of regional solutions, use of different communication systems, and the provision of superfluous or noninteroperable information.

It was further acknowledged that the content of MSPs will be developed by different international organizations, and thus coordination among these organizations is a priority to ensure harmonization of scope, format, structure, display on board, and communication systems used to transmit information electronically. While the work on contents of MSPs is currently undertaken by the IALA, the IMO Secretariat considers that the HGDM (IMO/IHO Harmonization Group on Data Modelling) should be tasked to work on the harmonization as outlined above. This interpretation concurs that a "general guidance" should be developed but should not define the detailed content of a particular MSP or aim at harmonizing the service itself. This is the responsibility of relevant data and service providers [13].

Reference [14] reports on the outcome of an informal meeting of member states and international organizations acting as domain-coordinating bodies for the further development of descriptions of MSs in the context of e-Navigation, held at IALA Headquarters on 9 October 2019 (Table 1).

### 3.2. Responsible Service Providers

In each country, there are authorities responsible for providing information services (INSs). Table 1 below offers examples of authorities responsible in each case, which can be different between countries. Responsible authorities may require service providers to deliver operational services.

The following six sea areas have been preliminarily identified for the delivery of MSPs:

1. port areas and approaches;
2. coastal waters and confined or restricted areas;
3. open sea and open areas;
4. areas with offshore and/or infrastructure developments;

5.  polar areas;
6.  other remote areas.

**Table 1.** Maritime service portfolios with responsible service providers [9,15].

| Service No | Identified Services | Identified Responsible Service Provider |
|:---:|:---:|:---:|
| 1 | Vessel traffic service (VTS) information service (INS) | VTS authority |
| 2 | Navigational assistance service (NAS) | VTS authority |
| 3 | Traffic organization service (TOS) | VTS authority |
| 4 | Local port service (LPS) | Local port/harbor authority |
| 5 | Maritime safety information (MSI) service | National competent authority |
| 6 | Pilotage service | Pilotage authority/pilot organization |
| 7 | Tug service | National competent authority; local port/harbor authority; private tug service company |
| 8 | Vessel shore reporting | National competent authority and appointed service providers |
| 9 | Telemedical assistance service (TMAS) | National health organization/dedicated health organization |
| 10 | Maritime assistance service (MAS) | Coastal/port authority/organization |
| 11 | Nautical chart service | National hydrographic authority/organization |
| 12 | Nautical publications service | National hydrographic authority/organization |
| 13 | Ice navigation service | National competent authority organization |
| 14 | Meteorological INS | National meteorological authority public institutions |
| 15 | Real-time hydrographic and environmental INS | National hydrographic and meteorological authorities |
| 16 | Search and rescue (SAR) service | SAR (Search and Rescue) authorities |

## 4. MS 1—VTS INS

The submitting organization is the IALA; the coordinating bodies are the IMO and IALA [8].

### 4.1. Description of MS 1

IALA Guideline 1089 on Provision of Vessel Traffic Services (INS, TOS, and NAS) provides guidances on the deliveries of three different types of services provided by a VTS: INS, traffic organization service (TOS), and navigational assistance service (NAS).

According to Resolution A.857(20) on Guidelines for Vessel Traffic Services, an INS provided by a VTS is defined as "a service to ensure that essential information becomes available in time for onboard navigational decision-making". Resolution A.857(20) also states that "the information service is provided by broadcasting information at fixed times and intervals or when deemed necessary by the VTS or at the request of a vessel, and may include for example reports on the position, identity and intentions of other traffic, waterway conditions, weather, hazards, or any other factors that may influence the vessel's transit".

Examples of the types of information provided by a VTS to operate an INS are presented in Table 2.

### 4.2. Purpose

The purpose of this MS is to provide data in a digital format to support a VTS INS and to create a means to reduce administrative burden and information overload, reduce miscommunication due to external interference, simplify work procedures, promote sustainable shipping and increase navigational safety.

Information provided in a digital format could complement and/or replace verbal/voice communications. The steps to achieve this transition to digital information exchange may vary in different areas and for different types of vessels. Details about digital information exchange should be published by VTS authorities.

### 4.3. Operational Approach

The digitalization of information will diversify communication means between shore authorities and vessels and will affect VTS procedures regarding provision of information.

Not all vessels are capable of receiving information in a digital format. Provisions should, therefore, be made to ensure that less capable vessels are receiving information they require. VTSs should remain the primary contact with vessels for urgent and important messages and to ensure communications with mariners.

### 4.4. Relations to Other MSs

MS 1 has relationships with other MSs, making it affect VTSs. Examples may be different depending on coastal state arrangements.

Examples of information are presented in Table 3.

**Table 2.** Examples of the types of information that may be provided by a VTS to operate an INS (IALA Guideline 1089) [8,9].

| Type of Information | Examples |
| --- | --- |
| Navigational situations (including traffic and route information) | • Position, identity, destination of vessels and the intention of other traffic <br> • Amendments and changes in promulgated information concerning a VTS area such as boundaries, procedures, radio frequencies, reporting points, and the mandatory reporting of movements <br> • Limited manoeuvrability that may impose restrictions on the navigation of other vessels or any other potential hindrances <br> • Suspension or change of routes, etc. |
| Navigational warnings | • Dangerous wrecks, obstacles not otherwise promulgated, diving operations, vessels not under command, etc. |
| Meteorology | • Information that includes the speed and direction of the prevailing wind, the directions and heights of waves, visibility, atmospheric pressure, the formation of ice, etc. |
| Meteorological warnings | • Gale, storm, tsunami, restricted visibility, etc. |
| Hydrography | • Information that includes factors such as the stability of the seabed, sea depth, the accuracy of surveys, tidal ranges, tidal streams, and prevailing currents and swell |
| Electronic navigational aids | • The availability of electronic navigational aids such as GNSS (Global Navigation Satellite System), Loran, DGPS (Differential Global Positioning System), AIS (Automatic Identification System), and Racon |
| Other information | • Port information, pilot or tug request, cargo information, health condition, PSC (Port State Control), ISPS (International Ship and Port Facility Security), etc. |

**Table 3.** Relations to other maritime services [8].

| Service No. | Maritime Service | Examples of Information Related to MS 1 |
| --- | --- | --- |
| MS 2 | VTS NAS | Under development |
| MS 3 | TOS | Under development |
| MS 4 | Port support service (PSS) | Delays, obstruction, cargo operations, port availability and anchorage area in the port, ISPS state, and MARSEC level |
| MS 5 | MSI service | All information depending on the structure of an MSI |
| MS 6 | Pilotage service | Pilot orders and updates |
| MS 7 | Tug service | Tug orders and updates |
| MS 8 | Vessel shore reporting | Notification of arrival, dangerous cargo, etc. |
| MS 9 | TMAS | delays |
| MS 10 | MAS | Notifications, routeing, and places of refuge |
| MS 11 | Nautical chart service | Local area updates and chart updates |
| MS 12 | Nautical publications service | Updates of publications |
| MS 13 | Ice navigation service | Ice routes, ice conditions, and ice-breaking assistance |
| MS 14 | Meteorological information service | Under development |
| MS 15 | Real-time hydrographic and environmental information services | Horizontal and vertical tidal information in a VTS area and available water columns |
| MS 16 | SAR service | Search pattern and vessel of opportunity |

## 5. MS 6—Pilotage Service

The submitting organization is the IMPA; the coordinating bodies are the IMO and IMPA [8].

### 5.1. Description of MS 6

Ships proceeding or leaving a port or a specific area should have easy access to information regarding the pilotage service provided. Information, such as local regulations, contact, notices,

means of boarding, boarding point, limitations, or pilot booking procedures, could be accessible by electronic means, where available.

The information provided through this service is not piloting information, as pilotage is a service physically performed on board ships by duly qualified and certificated or licensed maritime pilots.

### 5.2. Purpose—Information to Be Provided

This MS is limited to information provided to ships regarding a pilotage service in a given geographic area. It does not address the act of piloting, which is provided by a pilot on the bridge of a ship.

The purpose of this MS is to provide information related to a pilotage service when planning an operation before a pilot boards a vessel by using modern technology and common standards.

### 5.3. Operational Approach

Pilot organizations providing pilotage services in an area could provide information to ships about the pilotage services in a digital and easy accessible way. The information could be, as an example, portrayed as a layer on the ECDIS or in a graphical display. This information could include, for inbound ships, the location(s) of pilot station(s) or boarding point(s) in latitude/longitude or distances and bearings from a location, or an aid to navigation. In addition, the transmitted information could include VHF channels to contact the pilot or pilot boat. Typically, the pilotage service information will not be provided by the pilot, but rather by the pilot organization, because the pilot must be engaged in the actual performance of his/her pilotage duties.

In Reference [14], it is recalled that the initial description of MS 6 was drafted by the International Maritime Pilots' Association (IMPA). The IMPA indicated that it was satisfied with the level of description provided so far and that no further work was required. The IMPA expressed its concern that trying to harmonize pilot booking services through MSs, as discussed in the IMO Facilitation Committee (FAL) Correspondence Group, might not be consistent with pilotage services around the world.

Examples of information are presented in Table 4.

**Table 4.** Examples of information in MS 6 [8].

| Type of Information | Examples |
|---|---|
| General information | Examples of information: <br> • pilot requirements in an area; <br> • local regulations; <br> • limitations; <br> • requirements and procedures for ordering a pilot; <br> • requirements and procedures for pilot boarding; <br> • contact information of pilot stations; <br> • mandatory needs for tug assistance; and <br> • pilot boarding point. |
| Operational information | Examples of information: <br> • contacts for a pilot boat, launch, and a pilot helicopter; <br> • positions of a pilot station and a pilot boat; <br> • required arrangements for pilot boarding; <br> • boarding speed; <br> • communication; <br> • setup of a ship's radar, ECDIS (Electronic Chart Display and Information System), and other equipment as requested for the pilot's use; and <br> • any other actions requested of the ship for the pilot's benefit. |

*5.4. User Needs and Relations to Other MSs*

Ships are concerned by this service and need to know the pilot boarding/disembarking position, the pilot request procedures, local and special regulations, and the compulsory use of tugs.

MS 6 has relationships with other MSs, so that it affects the pilot boarding operation and contributes to safe and efficient operations.

## 6. MS 7—Tug Service

The submitting organization is Norway; the coordinating bodies are the IMO and Norway [8].

*6.1. Description of MS 7*

This MS ranges from small vessels with limited capacity and service in ports and rivers to ocean-going vessels built for complex operations and salvage. This service contributes to the safety of navigation, protection of the marine environment, and efficiency of marine transportation by conducting different types of operations, such as:

- transportation (personnel and staff between ports and anchorages);
- ship assistance (e.g., mooring);
- salvage (grounded ships or structures);
- shore;
- towage (harbour/ocean);
- escort; and
- oil spill response.

The needs of tug services differ from port to port and varied for different types of vessel and cargo. In some cases, information about a tug service capacity and/or availability may be difficult to obtain due to communication deficiencies. This MS is intended to improve information availability of tug services.

Tug services would encompass all kinds of tugs, such as:

- conventional tugs;
- azimuth stern drives;
- tractors; and
- rotors.

Examples of information are presented in Table 5.

*6.2. Purpose—Information to Be Exchanged*

This MS aims to facilitate access to all necessary tug-related information required by ships heading to ports, in order to optimize transit times and promote efficient movements of goods and persons by using modern technology and common standards.

Effective communications and exchanges of information between relevant stakeholders would contribute to efficient tug services. Electronic exchange of information would significantly contribute to the improvement of this service. For example, notifying a ship officer in advance about tug availability in a port could lead the ship to adapt its speed accordingly. In some cases, this may prevent a requirement to anchor the ship. The types of information, which can be exchanged, include:

- ETA (Estimated Time of Arrival) request;
- confirmation requests;
- updates on transit status and tug availability;
- updates among stakeholders; and
- standardized messages to overcome language barriers.

**Table 5.** Examples of information shared in a tug service [8].

| Type of Information | Examples of Information Shared in a Tug Service |
|---|---|
| Deep sea information | Examples of information: <br>• Contact information for tug vessels/operators <br>• Safety procedures and regulations <br>• Available resources <br>• Working hours |
| Local port or river information | Examples of information: <br>• Contact information for tug vessels/operators <br>• Mooring and berthing information <br>• Available resources <br>• Working hours |
| Tug information | Examples of information: <br>• Type of tug <br>• Capacity such as bollard pull <br>• Size <br>• Assistance services <br>• Response time <br>• Contact information <br>• Working hours |

*6.3. Operational Approach*

Tugs operations are a key element of the marine transportation chain, and well-coordinated procedures and communication means should be in place to ensure fluid movement of ships.

Like the port support service, this service can be significantly improved by utilization of a common platform to exchange information electronically and keep users updated on a regular basis about the status of operations, for both ships' operators and tug owners. The tug service aims mainly to improve communications involved in a ship request, rather than altering current operational procedures. Some of these communications may include:

- ship's size;
- number of tugs required;
- time, when the service is required;
- time, when the tug may be on site;
- estimated duration of operations; and
- end of operations.

Access to this information electronically would enhance the awareness of a ship's timestamp. Increased connectivity, through sharing of harmonized digital information regarding tug operations in ports, rivers, or deep sea, will enhance efficiency through just-in-time services. It will also reduce human factor errors, such as language barriers or outdated information in publications, enhancing efficiency and access to information in a fast and easy-to-use manner.

Example of an electronic communication platform for all actors involved in tug operations is illustrated in Figure 1.

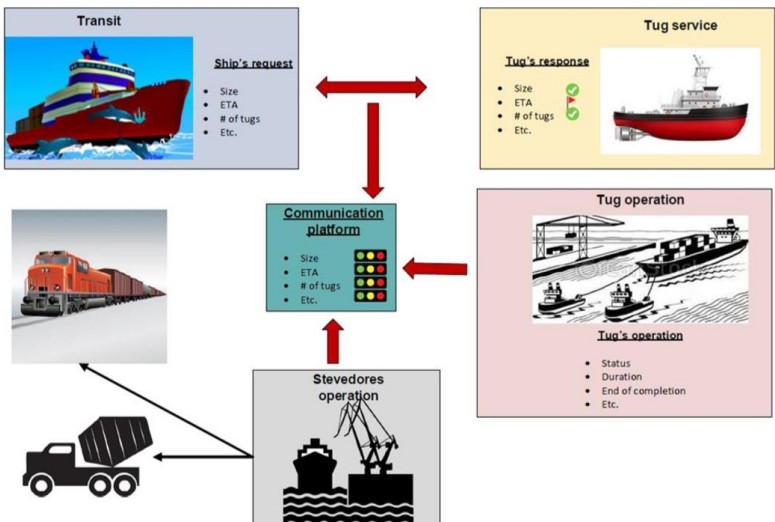

**Figure 1.** Example of an electronic communication platform for all actors involved in tug operations [8].

## 6.4. User Needs

Easy and timely access to tug service information is crucial to ensure fluidity in the transportation chain. The information required from this service is mainly related to:

- capacity;
- availability;
- time of response;
- status of operations; and
- durations of operations.

In return, tug services should be regularly updated on the ship's ETA/ATA (Estimated Time of Arrival/Actual Time of Arrival) to plan their operations accordingly. In the event of an unanticipated change, ship officers should be able to communicate tug services easily with each other to keep both parties informed about the evolving situation and allow for proper decision-making. An easy communication link should be part of user needs, and this communication link would also benefit all other actors.

## 6.5. Relations to Other MSs

Relations to other MS are presented in Table 6.

**Table 6.** Relation to other maritime services [8].

| Maritime Service | Examples of Information Related to MS 7 |
| --- | --- |
| MS 1—VTS INS | VTS area, types of VTS services, VTS contact information, places of refuge, local regulations, limitation, visibility, and information regarding the traffic in an area |
| MS 3—TOS | Traffic clearance and time slots |
| MS 4—PSS | Berthing information, time slots, security, local regulations, supply, assistance, and port contact information |
| MS 5—MSI service | Navigational warnings, meteorological information, and other urgent safety-related information |
| MS 6—pilotage service | Pilot regulations, contact information, and request procedures |
| MS 8—vessel shore reporting | Vessel information, cargo information, and crew information |
| MS 10—MAS | Contact information and places of refuge |
| MS 11—nautical chart service | Charting information and chart updates |
| MS 12—nautical publications service | Digital information from nautical publications that is relevant for operations at hand |
| MS 13—ice navigation service | Ice chart, ice conditions, information regarding icebreaker service/assistance, and ice routes |
| MS 14—meteorological information service | Information regarding the weather in an area |
| MS 15—real-time hydrographic and environmental information services | Information from real-time sensors |
| MS 16—SAR service | Salvage information, drifting parts, SAR areas, and rescue capabilities in an area |

## 7. Conclusions

This paper supports the IMO guidance on the definition and harmonization of the formats and structures of MSs in the context of e-Navigation and decisions to consolidate the descriptions of MSs and consider them together with all involved international organizations and interested member states, in order to harmonize the provision and exchange of maritime information and data. The paper strongly supports the initial descriptions of MSs in the context of e-Navigation approved by the MSC. The information contained in this paper constitutes the descriptions of two selected examples of MSs, an initial contribution to the harmonization of the formats and structures of pilotage and tug services.

This document shows a reasonable understanding of new developments in new technologies and e-Navigation for the maritime industry. The work has a good theoretical base, using useful references and important and updated IMO resolutions. It presents and studies the operational use of the e-Navigation concept in terms of requirements, implementation plans, and potential benefits/limitations in the maritime sector. It is possible to remotely operate several ships from land and over large geographical areas. The technology is used in different ways on vessels to show that the solutions can be applied widely. The marine scientific and engineering community aims to test and further develop key technologies linked to fully autonomous navigation systems, intelligent machinery systems, self-diagnostics, prognostics, and operation scheduling, as well as communication technologies enabling a prominent level of cybersecurity and integrating vessels into upgraded e-infrastructures.

Short descriptions of pilotage and tug services are presented in Table 7.

**Table 7.** Initial descriptions of pilotage and tug services [8].

| No. | Identified Services | Identified Service Provider | Short Description |
|---|---|---|---|
| MS 6 | Pilotage service | Pilot authority/pilot organization | The aim of the pilotage service is to safeguard traffic at sea and protect the environment by ensuring that vessels operating in a pilotage area have navigators with adequate qualifications for safe navigation. Each pilotage area needs highly specialized experience and local knowledge on the part of the pilot. Efficient pilotage depends, among other things, upon the effectiveness of communications and information exchanges between the pilot, the master, and the bridge personnel and upon the mutual understanding each has for the functions and duties of the officer. The pilot's portable unit (PPU) is a useful tool for safe navigation in clear and restricted visibility. Data accessible by the PPU should be made available in a structured, harmonized and reliable manner, and the interface for accessing such e-Navigation information should be standardized. Establishment of effective coordination between the pilot, the master, and the bridge personnel, taking due account of the ship's systems and equipment available to the pilot, will aid a safe and expeditious passage (see IMO Resolution A.960(23)). |
| MS 7 | Tug service | Port/commercial tug organization | Efficient tug operations depend on, among other things, the effectiveness of communications and information exchanges between relevant stakeholders. The aim of tug services is to safeguard traffic at sea and protect the environment by conducting operations such as:<br>• transportation (personnel and staff from port to anchorage) operations;<br>• ship assistance (e.g., mooring) operations;<br>• salvage (grounded ships or structures) operations;<br>• shore operations,<br>• towage (harbour/ocean) operations;<br>• escort operations; and<br>• oil spill response operations. |

In addition, the information service MS 1—VTS INS was also described as a template of what the initial description of the service should look like. The initial description of each of MSs is expected to be next periodically updated, taking into account developments and related work on harmonization.

All opportunities should now be used to promote the existence of the guidance and encourage those involved in MSs activities to implement the content of the guidance and, in the meantime, to expand the initial descriptions provided.

In conclusion, the following points should be noted:

– Emergence of new technologies and development of autonomous ships are key nowadays in the maritime industry with great contribution of the IMO;

- The paper tries to explain two important IMO resolutions for e-Navigation guidance, developments, and implementation;
- The paper provides a great example for tug services—key elements in marine transportation near ports, of which well-coordinated procedures assure fluid movements of ships and goods;
- Modern ships most probably will use less crew, but they will have high responsibilities for safe and efficient navigation;
- There is a development of better data exchanges and communications from ship to ship and ship to shore;
- Tools are created to have navigations and communications more reliable and minimize human errors, especially those with a potential for loss of life, injuries, maritime collisions, oil spills, environmental damage, and commercial costs;
- Automation and autonomous technologies and next-generation autonomous ships have been tested;
- The concept of e-Navigation was developed for remote-operated and autonomous maritime transport;
- Shipboard systems for autonomous navigation, self-diagnostics, prognostics, operation scheduling, and telecommunications are being developed, including digital MSs.

**Funding:** This study was financed by the Gdynia Maritime University, the research project: WN/2019/PZ/01.

**Conflicts of Interest:** The author declares no conflict of interest.

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
