# Peer review of "Initial Description of Pilotage and Tug Services in the Context of e-Navigation"

_jmse, doi:10.3390/jmse8020116_

Round 1

Reviewer 1 Report

More than 50% of this paper is directly copied from the IMO MSC 1/Circ. 1610. Although this document is referenced several times in the text it is not clear that so much of the text is actually a direct copy. This must be clarified. This could be done, for instance, graphically by using italics or a different font size

It must be made clear even if the author has written or taken part in writing the initial IMO document or the IMO/IALA documents that Circ 1610 are based on.

The development of the Maritime Service Portfolio is necessary for succeeding with e-Navigation. But one may question why the author feels it is necessary to write a paper supporting something that is already published by the IMO? But maybe political reasons make such a support by an academic paper useful. Which should then be acknowledged.

Otherwise the authors contribution to the paper is well written and offers a clear support to services exemplified.

Author Response

Thank you for your review.

I fully agree with your comments.

Reviewer 2 Report

- The article's main concept(s)

The author was a member of the IMO (International Maritime Organization) expert group that developed documents regarding Guidance on the definition and harmonization of the format and structure of Maritime Services in the context of e-Navigation (IMO MSC.467(101)). It was also involved in the IMO MSC.1/Circ.1610 regarding the Initial Descriptions of Maritime Services in the Context of e -Navigation.

The present times are introducing fast technology developments/implementations in the shipping industry. Concepts like e-Navigation are focused on better and more comprehensive support of the human operators. The automation, the modernization with information and communication systems are the future in ship operation.

Several companies are already talking about unmanned and autonomous ships. The technology contributes to safer, more efficient and sustainable operations at sea, both in containers and human transportation.

Considering that most maritime accidents are due to human errors, one of the actions is to implement e-Navigation to reduce human-based sea accidents. e-Navigation intends to enhance the navigation and related services for safety and security at sea and protection of the marine environment. The development of new technologies and its implementation for e-Navigation, can implement systematic maritime traffic management and therefore reduce marine accidents.

- Overall Comment

In overall,

 This document shows a reasonable understanding of new developments in new technologies and e-Navigation for the maritime industry. The work has a good theoretical base, using useful references and important and updated IMO resolutions.

It presents and studies the operational use of e-Navigation concept in terms of requirements, implementation plans and potential benefits/limitations in the maritime sector.

It is possible to remotely operate several ships from land and over large geographical areas. The technology is used in different ways on the vessel to show that the solutions can be applied widely.

Marine scientific and engineering community aim to test and further develop key technology linked to fully autonomous navigation systems, intelligent machinery systems, self-diagnostics, prognostics and operation scheduling, as well as communication technology enabling a prominent level of cybersecurity and integrating the vessels into upgraded e-infrastructure.

The manuscript has a simple structure, and it is easy to read/understand. It is journal paper but with an informative contribution regarding IMO e-Navigations guidance and field application of pilotage and tug services. The text needs some typos/errors revision.

The conclusions resume the paper contribution and explain common sense knowledge.

Some regards/remarks:

In line 35 – “when unmanned vessels will operate together with unmanned” – does the author mean that “Manned vessels will operate together with Unmanned vessels”?

There is a jump from section 6 to 8. Missing number 7 – minor typo.

- Weak and Strong points

Strengths

Great resume of two important IMO resolutions for e-Navigation guidance, developments and implementation; Great example with tug services – key element in marine transportation near ports – well-coordinated procedures assure fluid movements of ships and goods; Modern ships would use less crew – but they will have high responsibilities for safe and efficient navigation; Development of better data exchange and communications between ship to ship and ship to shore; Create tools to have navigation and communications more reliable and minimize human errors − especially those with a potential for loss of life, injuries, maritime collisions, oil spills, environmental damage and commercial costs; New technologies and the rise of autonomous ships are a key issue nowadays in the maritime industry – a great contribution; Teste automation and autonomous technology - next-generation autonomous ships; E-Navigation for remote-operated and autonomous maritime transport; Shipboard systems for autonomous navigation, self-diagnostics, prognostics, operation scheduling and telecommunications; …

Weakness

Very simple work – no real tests or application in a real scenario – just some proof of concept; Some minor typos, errors that need to be reviewed; …

Author Response

Thank tou for your review

I fully agree with your comments, advices and suggestions.